# Macrofungi Cultivation in Shady Forest Areas Significantly Increases Microbiome Diversity, Abundance and Functional Capacity in Soil Furrows

**DOI:** 10.3390/jof7090775

**Published:** 2021-09-18

**Authors:** Dong Liu, Yanliang Wang, Peng Zhang, Fuqiang Yu, Jesús Perez-Moreno

**Affiliations:** 1The Germplasm Bank of Wild Species, Yunnan Key Laboratory for Fungal Diversity and Green Development, Kunming Institute of Botany, Chinese Academy of Sciences, Kunming 650201, China; liudongc@mail.kib.ac.cn (D.L.); wangyanliang@mail.kib.ac.cn (Y.W.); zhangpeng@mail.kib.ac.cn (P.Z.); 2Colegio de Postgraduados, Campus Montecillo, Microbiología, Edafología, Texcoco 56230, Mexico

**Keywords:** macrofungi cultivation, fungal diversity, microbiome, quantitative real-time PCR, potential ecosystem function

## Abstract

Cultivating macrofungi is an important management measure to develop economy in shady forest areas; however, its effect on soil ecology, especially microbial abundance and structure, remains insufficiently studied. Herein, in a subtropical forestland, soil chemical and enzyme analyses, metagenomic sequencing and quantitative real-time PCR were employed to evaluate the impact of *Stropharia rugosoannulata* cultivation on soil microbiomes in three niches: soil below fungal beds, soil from furrows, and control forest soil with no influence from mushroom cultivation. Nutrients were accumulated in the soil below fungal beds with a significant increase (*p* < 0.05) in SOC, total C, total N, available P, and the activities of glucosidase and cellobiosidase. Non-metric multidimensional scaling and PERMANOVA results indicated that the structure of the microbiomes had been significantly (*p* < 0.05) shaped among the different niches. Soil furrows were microbial hotspots characterized by the higher microbial diversity and richness. Moreover, the increased microbiome abundance (assessed through qPCR) and the high number of significant stimulated functional types (based on MetaCyc genome database) indicated an enhanced functional capacity in furrows. Together, these results provide a comprehensive understanding of the microbial assemblies and the differently influenced soil properties in mushroom cultivation areas.

## 1. Introduction

Macrofungi, including members of phylum Basidiomycota and phylum Ascomycota in the kingdom of Fungi, have morphologically diverse epigeous or hypogeous fruiting bodies and are collectively referred to as mushrooms [1,2]. The wine-cap mushroom (*Stropharia rugosoannulata*) is one of the top ten mushrooms traded internationally and is recommended by FAO for export to developing countries [3,4]. High nutritional compounds (such as crude protein, crude fat, amino acids, minerals and vitamins) and bioactive compounds including antioxidant, anticarcinogenic and antidiabetes have been identified in *Stropharia* sporomes [5]. It is easy to cultivate wine-cap mushroom with a high yield under extensive management. Due to the economic (remote countryside development and food security) and ecological (the maintenance of forest masses and litter) benefits [6], the cultivation of *S. rugosoannulata* in forestlands has been vigorously promoted in the east [7], northwest [8] and southwest [9] regions of China. 

Recent studies have shown that cultivating macrofungi in shady forest areas can improve soil aeration, maintain soil structure [10], balance soil nutrient [11,12], increase soil biological activity [13] and shape bacterial taxa via hyphae expansion [14]. Nevertheless, in terms of forest management, the understanding of how the cultivation of *S. rugosoannulata* mushroom under subtropical forest modifies the soil physicochemical properties, the soil microbiome structure, and the functionality in different microniches remains limited.

The process of macrofungi cultivation involves artificial management strategies including forest land topsoil tillage, fermentation of mixed forest by-products (used as mushroom-cultivating substrates), inoculation of the targeted hyphae into the substrates (termed as “fungal bed”), the construction of furrows adjacent to fungal beds (used as paths for mushroom collection), and micro-spray system installment in forest canopies [7,9]. The varied soil moisture regimes (induced by artificial spray) could influence deeper soil organic carbon (SOC) decomposition rates and stability by mediating soil enzyme activities [15]. In general, soil carbon (C) components are decomposed by different soil extracellular enzymes. Labile carbon (LC) components (such as monosaccharides, starch, cellulose, and hemicellulose) are mainly decomposed by C-hydrolyzing enzymes, such as β-glucosidase, α-cellulase and β-xylosidase. Recalcitrant carbon (RC) components (i.e., lignin) are broken down mainly by oxidative enzymes such as peroxidases [16]. To understand the decomposition and retention mechanism of different SOC components in the macrofungal cultivation areas, soil nutrient and enzyme activities (especially oxidative enzymes due to presence of basidiomycetes macrofungi) needs to be further investigated. With the consideration of regular water-spray and moisture-stimulated SOC decomposition, nutrients from fungal bed can be leached into forest subsoils; therefore, we hypothesized that the main nutrients and C-decomposing enzyme activities of soils beneath the fungal bed would be substantially higher than those from furrows and control soils (sampled from non-cultivated adjacent forest areas; H1). Moreover, at the *S. rugosoannulata* cultivation plots, the furrows account for approximately half of the total cultivating area. Therefore, based on the ternary effects of soil water movement, nutrient-exchange and *S. rugosoannulata* hyphae expansion [17], it would be of great interest to understand how *S. rugosoannulata* cultivation affects microbiome assemblies and soil properties within different microniches. To better comprehend the influence of *S. rugosoannulata* cultivation on its surrounding soil microbiome, we employed high-throughput sequencing to investigate the diversity of soil bacterial and fungal community in mushroom-cultivating areas. Meanwhile, in order to evaluate the changes of microbial abundances, we performed a quantitative real-time PCR to quantify gene copy numbers of soil bacteria and fungi. In addition, soil microbiome potential metabolism profiles were predicted using a database of reference genomes [18,19]. It was hypothesized that microbiome diversity, abundance and putative function profile would be strongly affected in the different mushroom cultivation areas (H2). Furthermore, based on the expected variations in soil nutrients and microbiome assemblies in the different microniches, we expected that their relationships (which have important impacts on biochemical cycling in forest ecosystem) would also be differentially affected (H3).

## 2. Materials and Methods 

### 2.1. Site Description and Sampling Method

The experimental forestland was 6666 m^2^ in Dongshan Town, Mile City, Yunnan Province (103°67 E, 24°28 N, 2010 m above sea level). It is a state-owned, 30-year-old artificial *Pinus armandii* forestland. Trees were planted with 1 m space between plants and 1.5 m between rows. The region is characterized by a subtropical climate with a mean annual temperature and precipitation of 18.8 °C and 990 mm, respectively. Mushroom cultivation began on December 2018. The cultivating substrates (pine needles and branches) were collected locally within forest sites, and a pulverizer machine was used to crush them into fine pieces (with a width of 0.5 cm and a length of 5–8 cm). Crushed substrates were mixed for fermentation stacking. Then, the substrates were placed into the forest plots between tree row blocks approximately 1 m in length and inoculated into the fermented material as fungal beds. Fungal beds were firstly covered with 3 cm forest surface soil, and then a white plastic film. A micro-spray system was installed to maintain favorable moisture at the mushroom cultivating areas. Mushroom fruiting season was from March to June 2019. 

### 2.2. Soil Sampling and Chemical Analysis

All soil samples were collected on 28 March 2019. Four sampling plots (50 × 50 m) containing cultivated and non-cultivated areas, separated by transects of at least 50 m, were selected. In each sampling plot, three replicate soil samples were collected with dif-ferent influence of *S. rugosoannulata* cultivation: C = control, or soil without influence by mushroom cultivation located in nearby forestlands; L = soil with low influence by mushroom cultivation located in the furrows; and S = soil strongly influenced by mush-room cultivation located below the fungal beds (as shown in Figure 1). To minimize the effects of forest soil spatial variability, each of the three soil samples collected at each sampling plot consisted of three individual soil cores (10 cm height and 5 cm diameter), which were mixed together to form a single composite sample (as schematically shown in Figure 1c). Therefore, in total, twelve samples (4 sites × 3 single composite samples) were collected to be analyzed. Fresh samples were sealed in plastic bags and transported to la-boratory in iceboxes within 24 h. Half of each soil sample was sieved (<2 mm) to remove discernible roots, stones and macro-fauna and later air dried to measure the soil physico-chemical properties and soil acid phosphatase activity assay. The rest of the samples were stored at −20 °C for subsequent DNA extraction for high throughput sequencing. Backup samples were also stored at −20 °C. Standard methods for soil physicochemical parame-ters and enzyme activity analyses were used [20,21,22,23,24], and detailed descriptions of these protocols are provided in Appendix A.

### 2.3. DNA Extraction, PCR Amplification and High-Throughput Sequencing

Soil samples were extracted using the Power Soil DNA kit (12888, MoBio^®^, Carlsbad, CA, USA). Polymerase chain reaction (PCR) amplifications were carried out according to Joshi and Deshpande, (2011) [25]. The V4 hypervariable region of the bacterial 16S rRNA gene was amplified using the primer pairs 338F and 806R. For fungal communities, the internal transcribed spacer 1 (ITS 1) was used and the specific primers sets were ITS5F (5′-GGAAGTAAAAGTCGTAACAAGG-3′) and ITS1R (5′- GCTGCGTTCTTCATCGATGC-3′) [26]. PCR thermal cycling conditions were set under the following conditions: 98 °C for 2 min (initial denaturation), 30 cycles of 15 s at 98 °C, 55 °C 30 s, 72° C 30 s, and concluded with a final extension for 5 min at 72 °C. Amplicons were extracted from 2% agarose gels and purified with the Gel Extraction Kit (OMEGA bio-tek, Doraville, GA, USA) and were quantified on a Microplate reader (BioTek, Winooski, VT, USA, FLx800) using the dsDNA Assay Kit, Invitrogen (P7589, Carlsbad, CA, USA). Purified amplicons were pooled in equal amounts and pair-end sequenced 2 × 300 on the Illumina MiSeq platform, Miseq-PE250 (Personalbio^®^, Shanghai, China) using the MiSeq Reagent Kit v2 (600-cycles-PE, MS-102-3003). 

Sequences were processed using the QIIME 2 (Quantitative Insights into Microbial Ecology) pipeline following the steps of raw read quality control, paired-end clean read assembly, and raw tag quality control. Firstly, the exclusions of fuzzy base N and sequence lengths < 160 bp. Second, removal for the sequences with a mismatched base number > 1 of the 5′ end primer, and the sequences with >8 identical consecutive bases. Finally, deletion of chimeric sequences by filtering sequences on the USEARCH software (http://www.drive5.com/usearch/, accessed on 5 February 2019). To improve sequencing accuracy and avoid overestimation of bacterial diversity, singletons (sequences that occurred only once in dataset) were removed from downstream analyses. The obtained high-quality sequences were clustered into operational taxonomic units (OTUs) at a 97% similarity cutoff. OTUs taxonomic cluster was processed via searching reads against the Greengenes (for bacterial OTUs) [27] and UNITE (for fungal OTUs) database, respectively [28,29]; OTUs with abundance <0.001% were removed from final analysis [30]. The remaining OTUs were grouped according to their assigned taxonomic levels. All sequence data have been deposited to the ENA Sequence Read Archive under accession number SRP264748.

### 2.4. Quantitative Real-Time PCR

For quantifying gene copy numbers of soil bacteria and fungi, a quantitative real-time PCR (qPCR) was conducted using the 338F-806R for bacteria (10 μM each; [31]), and ITS3-ITS4 for fungi (10 μM each; [32]). The 20 μL qPCR reation mix contained 10 μL 2 × qPCRmix, 0.5 μL of each primer, 2 μL template DNA, 7 μL ddH_2_O. To estimate bacterial and fungal gene abundances, standard curves were generated using a 10-fold serial dilution of a plasmid containing a full-length copy of either the *Escherichia coli* 16S rRNA gene or the plasmid (STD102) for ITS2. Fluorescence intensities were detected in an Applied Biosystems StepOnePlus^TM^ Real-Time system Bio-Rad CFX96 Real-Time System (Thermo Fisher, Waltham, MA, USA) with the following cycling conditions: bacteria—95 °C for 5 min, 40 cycles of 15 s at 95 °C, 55 °C 30 s, 72° C 30 s, and a final melt curve of 60 to 95 °C; fungi—95 °C for 5 min, 40 cycles of 15 s at 95 °C, 60 °C 60 s, and a final melt curve of 60 to 95 °C. Three individual qPCR runs were performed for each replicate. Gene copy numbers were obtained from a regression equation for each assay relating the cycle threshold value to the known number of copies in the standards. 

### 2.5. Statistical Analysis

Microbial alpha-diversity (within community) was estimated by richness (Chao 1), diversity (Shannon and Simpson) and evenness (Pielou_e) indices. One-way analysis of variance (ANOVA) followed by Tukey HSD was used to compare significant differences in the diversity indices. Beta-diversity (between-habitat difference) was calculated using pairwise Bray-Curtis distances [33] and visualized using non-metric multidimensional scaling (NMDS) plots using Vegan package in R environment. Permutational multivariate analysis of variance (PERMANOVA) was used to verify the significant difference (overall and pairwise) in the treatments presented in NMDS plots. Linear discriminant analysis effect size (LEfSe) was used to identify microbial biomarker by a threshold of LDA score >2.0 and *p* < 0.05 [34] in bacterial and fungal communities. The cladogram was presented by LEfSe algorithm via online platform (http://huttenhower.sph.harvard.edu/galaxy, accessed on 15 February 2020). To obtain a deeper insight into the main soil parameters that could explain the differences in microbial communities in the three evaluated treatments, and also whether these parameters were the same between bacterial and fungal communities, we selected redundancy analysis (RDA) or canonical correspondence analysis (CCA) based on the gradient length by detrended correspondence analysis ordination axis (<3, RDA; 3–4, CCA) [35,36]. For the bacterial community, the response data were compositional and had a gradient of 1.4 SD units, so the linear method (constrained RDA) was used. For the fungal community, response data had a gradient of 3.6 SD units, so a unimodal method (CCA) was selected. Variation partitioning analysis (VPA) was used to quantitatively elucidate the impact of soil variables on microbial communities. Prior to VPA, a subset of soil parameters (moisture, AN, AFe, Ca^2+^ and Mg^2+^) having significant relationship (*p* < 0.05) with the matrix of soil microbial communities were selected following the CCA/RDA analyses. Then, the obtained data were used for subsequent VPA in Canoco 5.0 statistical software.

### 2.6. Microbial Putative Functional Profile Analysis

The PICRUSt2 (Phylogenetic Investigation of Communities by Reconstruction of Unobserved States 2) software was used to predict the functional abundance by using the abundance of the tagged gene sequence in samples (Gavin m. Douglas et al., preprint). Based on the reference genome data of the software, the 16S rRNA sequence and ITS sequence can both be used for functional prediction. The reference genome database of PICRUSt2 is 10 times larger than the original version PICRUSt (Langille et al., 2013) [18] and with higher numbers of MetaCyc metabolic pathways (https://metacyc.org/, accessed on 20 May 2020). MetaCyc is the largest metabolic reference database in the field of life sciences that has been elucidated by experimental data. Currently, it contains 2722 pathways from 3009 different organisms [37]. Detailed analysis process of the PICRUSt2 can be found in https://github.com/picrust/picrust2/wiki, accessed on 2 July 2020.

## 3. Results

### 3.1. Cultivation of Macrofungi Changes Soil Properties

The cultivation of *S. rugosoannulata* in forestlands generally increased the soil pH, organic matter, total C, total N, total P, alkaline hydrolysable N and available P contents, as well as Ca^2+^, Mg^2+^, available Mn and available Fe concentrations (Table 1 and Table 2). All the measured soil properties increased significantly in the strong influenced area (S) compared with the control soil (C) except for total P (Table 1 and Table 2). Generally, there were no significant differences between the control and the less influenced areas (L) (*p >* 0.05). Although total P content showed a minor variation among treatments, significant differences (*p <* 0.05) were found in the available P and acid phosphatase activity among soils, following the order S > L > C (Table 2 and Table 3). The activities of two hydrolases (glucosidase and cellobiosidase) were significantly higher in S than in C and L, but the oxidase activity had no significant change among treatments (Table 3).

### 3.2. Various Distribution of Bacterial and Fungal Taxa and Phylotypes

Illumina MiSeq high-throughput sequencing analysis of 16S rDNA gene was conducted to determine the bacterial community structure. Raw reads obtained from different samples ranged from 30,310 to 54,713 per sample (mean = 42,511) with length of 235–440 bp. Rarefaction analysis was conducted on each sample and all of the bacterial rarefaction curves reached the plateau phase at the sequencing depth of 12,798, suggesting that all soils were sampled to saturation (Appendix A). The bacterial rarified observed richness ranged from 1524 to 1794, and was significantly higher in L (1763 ± 24) than in C (1584 ± 73) and S (1537 ± 111) (*p* < 0.05; Appendix A). A total of 121,257 high-quality V4–V5 16S rDNA sequences were analyzed. These sequences were assigned to 4044 OTUs (operational taxonomic units). The number of OTUs of individual samples ranged from 743 to 1341. The good coverage values varied from 99.26% to 99.84%, indicating that these sequences were sufficient to analyze the bacterial community structures. 

Major bacterial taxa (relative abundance > 10%) across all soil samples belonged to Proteobacteria (averaged 31%), Acidobacteria (22%), Chloroflexi (18%), and Actinobacteria (13%). These groups were responsible for more than 80% of the total bacterial sequences obtained (Appendix A). Groups of Gemmatimonadetes, Planctomycetes, Verrucomicrobia, and Bacteroidetes were less abundant (relative abundance > 1% and <10%), but were still identified in all soils. Substantial changes were observed at phylum level. The relative abundance of Chloroflexi strongly increased from 14% (in C) to 21% (in S), compared with a clear decreasing in the relative abundance of Actinobacteria from 19% in C to 8% in S (Appendix A). On the class level, Acidobacteria, Alphaproteobacteria, AD3 (within the phylum of Chloroflexi), and Gammaproteobacteria were dominant with a mean relative abundance of ~10%, followed by Gemmatimonadetes (8%) and Actinobacteria (7%). The classes of Thermoleophilia, Deltaproteobacteria, and Plantomycetacia were less abundant (~3%) among samples (Appendix A). At the genus level, there were 48 genera with a relative abundance >0.5%. Approximately 19 and 8 genera belonged to uncultured and unclassified bacteria, respectively. Among the rest of the 21 genera, the mean relative abundances of the top three genera (2~3%) were *Bradyrhizobium, Gemmatimonas* and *Acinetobacter.* Meanwhile, the *Acidibacter, Acidothermus, Bryobacter, Conexibacter, Mycobacterium* and “*Candidatus* Udaeobacter” were also observed in all soils, with mean relative abundances ~1% (Figure 2A).

The rarefaction curves of fungal OTUs reached the saturation at the sequencing depth of 15,389 (Appendix A). The fungal rarified observed richness ranged from 105 to 279, and was significantly higher in L (245 ± 24) than in C (167 ± 57) and S (155 ± 76) (*p* < 0.05; Appendix A). For fungal diversity, 255,341 high-quality ITS1 sequences were analyzed. The good coverage values reached a minimum value of 99.93%. The main fungal phyla (P) were Basidiomycota and Ascomycota (occupying > 97% of the fungal sequences, Appendix A). Approximately 1.2% (unclassified fungi) and 0.8% (unidentified) of the fungal sequences could not be clustered into definite taxa classes (Appendix A). At class level, Agaricomycetes (Phylum Basidiomycota) was most abundant (with a mean of ~69% sequences across all samples), followed by Tremellomycetes (~12%) (Phylum Basidiomycota), Eurotiomycetes (~5%) (Phylum Ascomycota) and Sordariomyetes (~4%) (Phylum Ascomycota). The classes Leotiomycetes and Pezizomycetes belonging to phylum Ascomycota were less abundant (relative abundance 2~3%) but were still present in all soils (Appendix A). Additionally, >200 fungal genera were detected (Appendix A). There were 23 fungal genera that had relative abundance over 0.5%, among which 6 genera remain unidentified/unclassified (Appendix A). Soil fungal communities were conspicuously dominated by distinct fungal genera in the different evaluated treatments. As expected, in S soil, *Stropharia* was dominant with a relative abundance of ~24%, followed by *Suillus* (21%), while in L soil, *Thelephora, Russula* and *Saitozyma* were evenly abundant with a relative abundance of 15%. Meanshile, *Suillus* (37%) and *Tricholoma* (30%) were predominated in the control soil (Figure 2B).

### 3.3. Bacterial but Not Fungal Diversity Showed Significant Changes

For bacteria, the richness index (Chao 1) was statistically higher in L than in C and S; diversity (Shannon and Simpson) and evenness (Pielou_e) indices were significantly higher in C and L, than in S (*p* < 0.05; Figure 3). For fungi, the Shannon, Simpson and Pielou_e indices did not differ among the *S. rugosoannulata* cultivation areas (C, L and S). Only Chao1 index in L was significantly higher than in C and S (*p* < 0.05; Figure 3). In summary, in terms of richness, both bacterial and fungal communities were higher in L than in C and S. However, in terms of diversity and evenness, C and L showed higher diversity than S for bacteria but there were no differences for fungal communities. 

### 3.4. Both Bacterial and Fungal Community Structure Showed Significant Changes 

To compare the beta-diversity of soil microbial communities, non-metric multidimensional analyses (NMDS) were performed for bacterial (Figure 4A) and fungal (Figure 4B) OTUs. For the bacterial community, NMDS distinguished the three different soil microniches (PERMANOVA test, *F* = 1.509, *p* = 0.032), whereas the difference between treatments L and S was not significant (PERMANOVA test, *F* = 0.842, *p* = 0.546). The difference of bacterial community structure increased from C, L to S (as shown by diverging of the shape; Figure 4A). Contrastingly, for fungi, communities showed no clear separation between L and S (PERMANOVA test, *F* =1.469, *p* = 0.140; Figure 4B). Both communities exhibited a distinction between the control (C) and mushroom cultivation areas (L and S), bacterial community exhibited stronger difference (stress = 0.0708) than that of fungal community (stress = 0.134). 

For bacterial and fungal communities, the Bray-Curtis distance matrix was significantly different between the control and mushroom cultivation areas (Table 4), indicating that the phylogenetic structure of both bacterial and fungal assemblies was affected by mushroom cultivation. Meanwhile, there were no statistical differences between the soil microbiome communities in L and S (Table 4).

Analyses of the core and unique bacterial and fungal OTUs in C, L and S, showed that the unique OTUs were much higher than the OTUs shared by all treatments in the case of bacteria (7% vs. 22 to 29% of the total number of OTUs) (Figure 5A,B). Meanwhile, an opposite trend was shown in the case of fungi, showing that mushroom cultivation affects differentially the structure of different microbial functional groups. LEfSe analysis identified 27 bacterial and 17 fungal biomarkers in C, L and S (Figure 5C,D). In soils with strong influence of *S. rugosoannulata* cultivation, there were six highly enriched bacterial taxa and four fungal taxa. Meanwhile in L, eight and nine bacterial and fungal taxa were highly abundant. Among them, Acidothermaceae and *Venturia* were the most obvious biomarkers having the highest LDA scores (Figure 5). In control soils, there were 13 bacterial and 4 fungal taxa identified as biomarkers, with Streptomycetales and Tricholomataceae being the most enriched taxa (Figure 5).

Quantification by qPCR resulted in high microbial abundances in all soils of different mushroom cultivation areas, amounting to 2.7 × 10^10^ and 1.8 × 10^9^ mean gene copy numbers per gram of soil for bacteria and fungi, respectively (Figure 6). The total microbial (bacterial and fungi) gene copies ranged from 8.6 × 10^8^ gene copies in C soils to 3.8 × 10^10^ gene copies in L soils. Significant differences in microbial abundances were observed among the three treatments, with the significantly highest values observed in the areas with low influenced area by mushroom cultivation (L), followed by the strongly influenced area by mushroom cultivation (S) and the control soil (C). This trend was the same for both for the number of total microbial gene copies and for bacteria and fungi, calculated separately. 

### 3.5. Microbiome Function Prediction

In total, there were 16 and 8 significantly changed functional types based on MetaCyc genome prediction from a total of 27 and 50 for bacterial and fungal community, respectively (Appendix A). For the bacterial community, *S. rugosoannulata* cultivation has induced significant changes of 12 putative functions, among them respiration and fatty acid and lipid degradation were highly enriched in the soil from furrows (Figure 7A); the C1 compound use and assimilation was significantly enriched in the soil below the fungal bed (strong influenced area; Figure 7A); when comparing between the less and strong influenced area, the functional prediction related to respiration, TCA cycle and glyoxylated cycle were stimulated in the furrows. However, for the fungal community, only eight putative functions significantly changed (Figure 7B) in *S. rugosoannulata* cultivation areas (Figure 7B). The nucleic acid processing, phospholipases, cofactor, prosthetic group and electron carrier degradation were significant abundant in the furrows and the soil below the fungal bed, while other putative metabolism of respiration, amine and polyamine degradation and various types of biosynthesis significantly decreased compared to the control soil outside the mushroom cultivating area (control, Figure 7B).

### 3.6. Correlation between Soil Properties and Microbial Communities

Redundancy analysis (RDA) or canonical correspondence analysis (CCA), analyzed as explained in detail in Materials and Methods section, showed clear relationships between some physical and chemical soil properties and specific microbial genera in the evaluated treatments (Figure 8). Several bacterial and fungal genera were strongly and differentially influenced by soil factors; however, in general, the variation in bacterial communities was mainly driven by soil moisture, two available minerals (Fe and N) and total Mg^2+^ and Ca^2+^ ions. For fungal communities, moisture, Ca^2+^ and Mg^2+^ ions were again the top three most influential variables, whereas available Fe and N did not have an influence as strong as in the case of bacterial genera (Figure 8B). 

Factors explaining differences in microbial communities in the three treatments were calculated by RDA/CCA. For the bacterial community, the eigenvalues were 0.59 and 0.15 for axis 1 and 2, which explained 58.9% and 14.7% of the total variation, respectively (Figure 8A). For the fungal community, the eigenvalues were 0.68 and 0.47 for axis 1 and 2, which explained 30.7% and 21.1% of the total variation, respectively (Figure 8B). Forward selection showed that soil alkaline hydrolysable nitrogen (AN), available Fe (AFe), soil moisture, the concentrations of calcium (Ca^2+^), and magnesium (Mg^2+^) ions were the primary soil variables (*p* < 0.05) for changes in soil bacterial communities (Figure 8). Interestingly, in the control soils, the *Bradyrhizobium* was negatively related with AN, Ca^2+^ and moisture, while in the mushroom cultivation soils, this correlation changed from negative to positive (Figure 9). The opposite occurred, as well, in the case of the relationship between Ca^2+^ and *Gemmatimonas*, which changed from positive (in the control) to negative (in the mushroom cultivation soils; Figure 9), showing clear relationship shifts depending on the evaluated microniches. The variation in fungal communities was mainly driven by soil moisture, Ca^2+^ and Mg^2+^ ions (*p* < 0.05) for the most abundant (top six in relative abundance; Appendix A) fungal genera: *Suillus*, *Saitozyma*, *Stropharia, Tricholoma*, *Thelephora* and *Russula*. This demonstrates that mushroom cultivation originates dramatic changes between the correlations of these fungal genera (especially for *Russula* and *Saitozyma*) and soil driving factors (moisture, Ca^2+^ and Mg^2+^ ions) (Figure 9). This was particularly dramatic in the case of the relationships between different enzymatic activities and some of the most abundant fungal genera in the furrows compared to C and S (Figure 9G–L). For example, there was a shift from negative to positive peroxidase production in the furrows compared to C and S treatment in the bacterial genera *Conexibacter, Mycobacterium, Granulicella* and *Rhodoplanes*, as well as in the fungal genera *Stropharia, Suillus, Trichopea* and *Tomentella*. A similar trend was recorded for glucosidase in the bacterial genus *Gemmatimonas* and in the fungal genera *Saitozyma, Sagenomella, Penicillium, Phialocephala* and *Nectriopsis.* Finally, the same phenomenon was recorded for cellobiosidase production in the bacterial genus *Conexibacter* and in the fungal genera *Saitozyma* and *Trichophaea* (Figure 9G–L). VPA was used to quantitatively elucidate the impact of soil moisture and available nutrients on soil microbial communities in *Stropharia rugosoannulata cultivation* areas (Figure 10). The combination of the selected soil parameters showed a significant correlation with the soil bacterial (Mantel test, *r* = 0.531, *p* = 0.001) and fungal community (Mantel test, *r* = 0.429, *p* = 0.007) structure. For the bacterial community structure, soil moisture and available nutrients explained 59% of the total variation. Among these, soil Ca^2+^, AFe, moisture, Mg^2+^ and AN explained 16.5, 15.8, 14.5, 8.5 and 4.2% variations in bacterial communities, respectively (Figure 10A). Meanwhile in the case of the fungal community structure, soil moisture and available nutrients explained 46.2% of the total variation. Soil moisture explained 21.6%, while soil Mg^2+^ and Ca^2+^, explained 13.5% and 11.1% variations in fungal communities, respectively (Figure 10B). 

## 4. Discussion

### 4.1. Effect of Macrofungi Cultivation on Soil Properties

The rationale of our work was to investigate how wine-cap mushroom (*Stropharia rugosoannulata*) cultivation increases the soil fertility in forest ecosystems. In line with our first hypothesis (H1), the SOC, total C and total N were significantly higher in the area strongly affected by mushroom cultivation (i.e., soil beneath mushroom cultivating fungal beds) than in soils from furrows and nearby the cultivation areas, which could be a consequence of increases in beneficial microorganisms such as N fixing or C fixing microbes. Compared to the non-cultivated soil, the relative abundances of Chloroflexi (Photosynthetic autotrophic bacteria; [38]) and Gemmatimonadetes (N-fixer) were increased almost twofold in the strongly influenced soil (from 14% to 21% for Chloroflexi; and from 5% to 9% for Gemmatimonadetes). Meanwhile, the soil beneath the fungal bed had significantly higher stoichiometry (C:N, C:P and N:P) ratios compared to soils from furrows and outside the cultivation areas (Table 1). The increased soil stoichiometry reflects a decreasing nutrient cycling rate [39,40], which benefits C, N and P fixation. In addition, the easily available nutrients (Ca, Mg, Mn and Fe) were also significantly increased in the soil beneath the fungal bed (Table 2). This could be related to (i) the boosted microbial mineralization in the fungal bed, because the C:N ratio substantially increased from 16 (fungal bed itself) to 49 (soil beneath the fungal bed; Table 1), reflecting a stimulated microbial-mediated fermentation (Holland 1992), increasing the conversion of organic (crushed pine needle and branch in fungal bed) to inorganic compounds; (ii) the micro-spray watering system leading to a significantly increased moisture benefiting the nutrients leaching from the fungal bed, increasing eluviation of mineral elements into the strongly influenced soil area [41,42]; and (iii) the acidic soil condition increasing acid phosphatase activity enzyme activity (Table 3), favoring the acid leaching process and increasing the amount of eluent substances (Fujii et al., 2009). 

We found that the activities of two hydrolases (glucosidase and cellobiosidase) were significantly boosted in the soil beneath the fungal bed as compared to the other areas, but the oxidase activity remained stable. The reasons for various responses might be the preferential use of labile C by microbes, for instance, after wine-cap mushroom cultivation in shady forest areas (with micro-spray irrigation system installment in the forest canopy), and soil microbes tending to use labile C components, and thus not increasing the input of oxidase for decomposing recalcitrant C in the forest soil beneath the cultivation beds [15], because this activity is concentrated in the cultivation beads themselves, where there is abundant organic matter. On the other hand, from an ecological point of view, in shady forest areas, cultivation of mushrooms showed a positive effect of the use of fungal beds, obtained as a subproduct from the same local forested areas, in forest soils, as it favors the stabilization of the soil recalcitrant C pool, which plays crucial roles in major soil functions and ecosystem services [17]. 

### 4.2. Effects of Macrofungi Cultivation on Soil Microbial Communities

With the consideration of the co-effects of nutrient leaching, *Stropharia*’s hyphae expansion, and the exchange/absorption of nutrients from adhering soils, we further hypothesized that the microbiome diversity, abundance and putative function would be strongly shaped in the mushroom cultivation areas (H2). In accordance with H2, microbial alpha-diversity (except for fungal diversity and evenness), the uniqueness of bacterial and fungal OTUs, microbial abundance, and the microbiome structure significantly changed in soils beneath the mushroom cultivating fungal bed, furrows and outside of the cultivation areas (Figure 3, Figure 4, Figure 5 and Figure 6). This indicates that the mushroom cultivation differentially shaped the microbial assemblies in the different niches, particularly with respect to L and S. 

Consistent with the changes in microbial diversity and abundance, the significantly different microbial putative functional profiles were all higher in soils from furrows than in soils below the fungal beds (Figure 7). Beneath the fungal bed, the significantly stimulated C1 compound use and assimilation was in line with the increased organic matter in this area (Table 1); however, in the furrows, putative function involved in bacterial soil respiration presented the highest functional profile, with a significant increase compared to C and S (Figure 7). Based on a functional prediction, our results suggested that *Stropharia* cultivation would increase the flux of CO_2_ in the soil next to fungal bed, while simultaneously increasing carbon sequestration in the soil beneath the fungal bed. In other words, the negative effect (CO_2_ emission) could be largely combated by the C stock from mushroom cultivation in forest soils. This is a carbon-pool practice in terms of forest ecology and management, especially when compared with the traditional tillage cultivation that stimulates soil CO_2_ flux without increasing SOC stock [43]. 

It should be noted that the change of microbiome was different from that of soil nutrients. Although soil nutrient contents were highest in soil beneath the fungal bed, soil microbial richness, evenness and diversity were all higher in soils next to the fungal bed than in those beneath the fungal bed (Figure 2). The inconsistent variation between the microbial indexes and soil nutrients can be explained by a ‘microbial growth lagging effect’ induced by improper resource stoichiometry [44,45], that is, soil nutrients were there, but were not available for microorganisms in soils below the fungal bed (significantly high C:N:P ratio; Table 1). These results indicate that mushroom cultivation directly alters the C:N, C:P, and N:P ratios of soil below the fungal bed, with lagging repercussions for microbial taxa.

### 4.3. Changes in the Relationship between Microbiome and Soil Properties

In accordance with our third hypothesis (H3), the mushroom cultivation resulted in dramatic changes between the correlations between microbial genera and soil properties (Figure 6). For example, the changes in environmental factors caused by *S. rugosoannulata* cultivation dramatically modified the environmental impact on microbial diversity, even causing a clear shift from positive to negative correlations (or vice versa) in some cases, e.g., the correlation between *Bradyrhizobium* and available N, Ca and moisture was changed from negative to positive (Figure 9), while the opposite occurred as well in the case of the correlation between Ca and *Gemmatimonas*, which changed from positive to negative (Figure 9). Quantification by qPCR showed that furrows presented higher bacterial and fungal microbial abundances than S and C treatments. Similarly, Spearman analysis revealed that relationships between enzymatic activities and the most abundant fungal and bacteria soil genera shifted from negative to positive in this soil areas when compared with S and C. These shifts recorded for different bacterial and fungal genera are of great ecological relevance due to the fact that they indicate that, in the furrows, usual ecological activity in the forest soil continues despite the mushroom cultivation. For example, it is well known that external ectomycorrhizal mycelium extracts nutrients from organic materials, such as leaf and root litter by peroxidase activities, which is a decisive factor in the regulation of soil C storage and mediates the response of ecosystem C sequestration [46]. In our study, a shift from negative to positive peroxidase production was recorded in the furrows compared to S and C treatments in genera that have been reported to be ectomycorrhizal, including *Suillus*, *Tomentella* and *Trichophaea* [47], therefore making it an indicator of C cycling. Similarly, glucosidase production was shifted in furrows of our evaluated system from negative to positive compared to S and C treatments in bacteria (*Gemmatimonas*) and fungi (*Saitozyma*, *Sagenomella*, *Penicillium*, *Phialocephala* and *Nectriopsis*). This enzyme is well known to transform starch to single sugars, and it has even been considered to be an indicator of soil C accumulation in soil forest ecosystems [48]. These facts have a paramount relevance because they demonstrate, for the first time, that along with their pragmatic use for mushroom cultivation in shady forest areas, furrows constitute an important genetic reservoir of natural forest soil microbiomes when the mushroom cultivations come to an end, and also that furrows are able to retain their biological functionality despite their closeness with the fungal beds in which mushrooms cultivation is being conducted. Previous complex variations in these relationships might be induced by the substantial variations in soil available nutrients and corresponding effects on soil bacteria [49,50], because in our case, the iron and calcium were the most influencing factors shaping soil bacterial microbiome in the *S. rugosoannulata* cultivation area. This could be closely related to the roles of bacteria in the biogeochemical cycles of the trace elements in forest soils [49]. Among the investigated trace elements, the strongest effect of calcium on soil bacterial communities was attributed to its roles in inhibiting the growth of pathogenic bacteria [51] and the synergetic effect of promoting the levels of magnesium and carbohydrate [52]. In comparison, in the same forest soil areas, the fungal communities were more strongly affected by the soil moisture (Figure 8 and Figure 10). In the present study, we found that the moisture was a major determinant of soil fungal community composition, which was in line with the findings in a similar acidic forest soil in northeast China [53] and in western Canada [54]. Moisture has been found to be a regulator of soil arbuscular mycorrhizal [55] and ectomycorrhizal fungal community assembly [56]. Moreover, as a physiological stress index, the strong influence of moisture on soil fungal community might also be associated with its mediation of dissolvable nutrients and microbial metabolic functions [57].

## 5. Conclusions

Our results showed that the effect of macrofungi (*Stropharia rugosoanulata)* cultivation was stronger in the soil below the fungal beds than the soil from the furrows with respect to soil overall nutrients and microbial communities. In the soil beneath the fungal beds, significantly increased macronutrients and easily available nutrients showed different influences on bacterial and fungal community structure. However, the improved soil conditions did not lead to an increase in soil bacterial and fungal diversities; this lagging effect ought to be associated with an inconsistent stoichiometry between soil and microbes. The soil furrows were shown to be a functional hotspot for soil microbes, as they were characterized by higher microbial richness and abundance, as well as a strongly active functional niche. This is the first time that this phenomenon has been recorded for mushroom cultivation in shady forest areas. Overall, our results highlight that mushroom cultivation in shady forest areas increases their soil fertility and strongly modifies the forest soil microbiome community structure.

## Figures and Tables

**Figure 1 jof-07-00775-f001:**
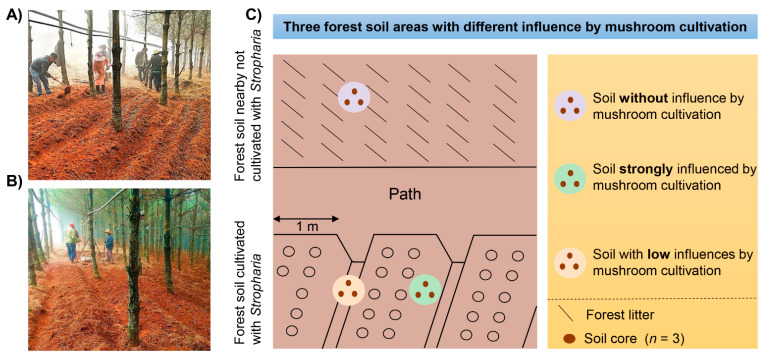
Overview of the forest sites (**A**,**B**); and sampling scheme (**C**) of *Stropharia rugosoannulata* cultivation areas: (i) Grey circle = Control forest soil with no influence from mushroom cultivation (C); (ii) Pale pink circle = Soil from the grooves with low influence by mushroom cultivation (L); and (iii) Green circle = Soil below the fungal beds strongly influenced by mushroom cultivation (S).

**Figure 2 jof-07-00775-f002:**
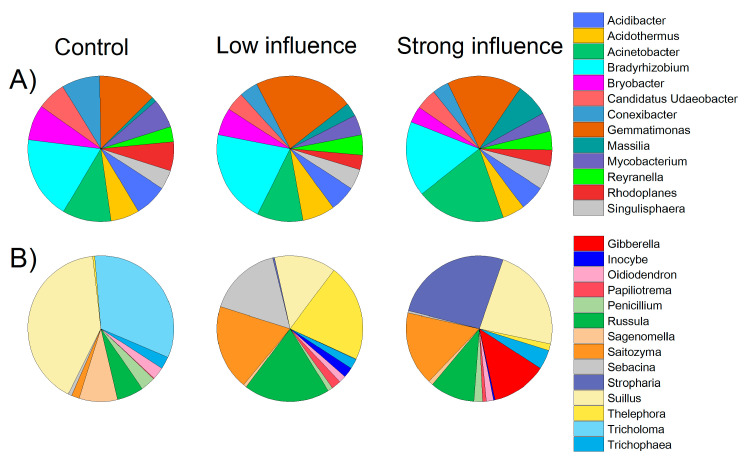
Abundant (>0.5% relative abundance) soil bacterial (**A**) and fungal (**B**) genera found in soils with low and strong influence of *Stropharia rugosoannulata* cultivation and control soils. Uncultured and unclassified/unidentified genera are not presented. Detailed information of the treatments is described in Table 1.

**Figure 3 jof-07-00775-f003:**
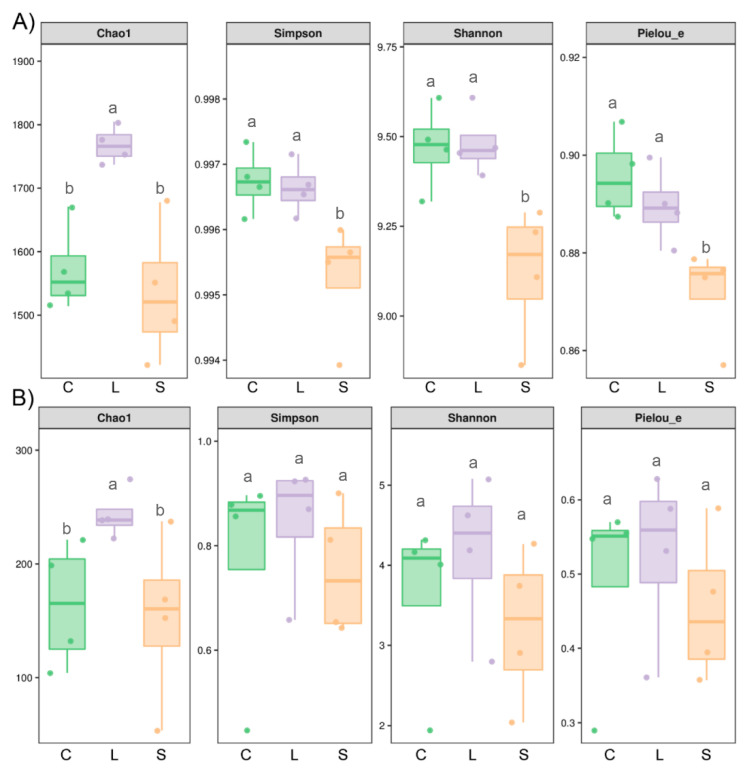
Soil bacterial (**A**) and fungal (**B**) diversity indices in soils with low (L) and strong (S) influence of *Stropharia rugosoannulata* cultivation and control (C) soils. Alpha diversity indices were based on microbial richness (Chao 1 index), diversity (Shannon and Simpson) and evenness (Pielou_e). For individual index boxes, different letters indicate significant differences at *p* = 0.05 (ANOVA) between means (Tukey’s HSD pairwise comparisons, *n* = 4).

**Figure 4 jof-07-00775-f004:**
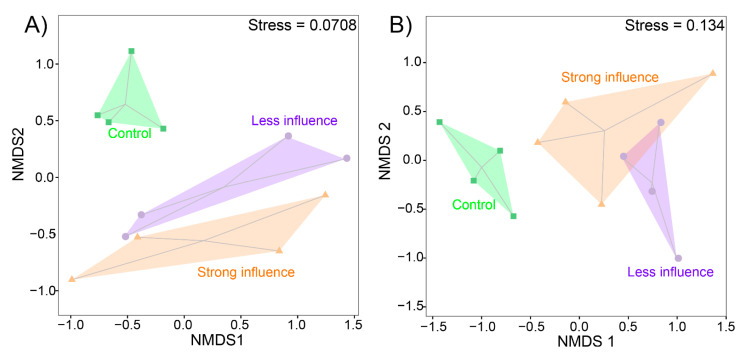
Bacterial (**A**) and fungal (**B**) community compositions as indicated by non-metric multi-dimensional scaling plots (NMDS) of pairwise Bray-Curtis distance in three different *Stropharia rugosoannulata* cultivation areas. Treatment descriptions are those described in Table 1.

**Figure 5 jof-07-00775-f005:**
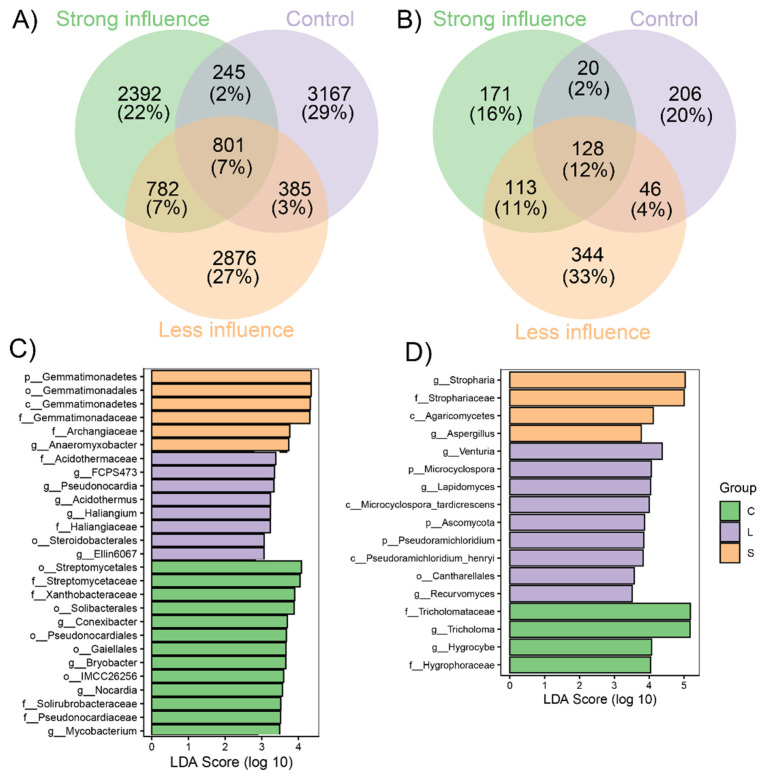
Core and unique bacterial (**A**) and fungal (**B**) OTUs in three different *Stropharia rugosoannulata* cultivation areas with various influencing intensities; and linear discriminant analysis (LDA) value distribution histogram of (**C**) bacteria and (**D**) fungal taxa. For C and D, p = phylum, c = class, o = order, f = family and g = genus.

**Figure 6 jof-07-00775-f006:**
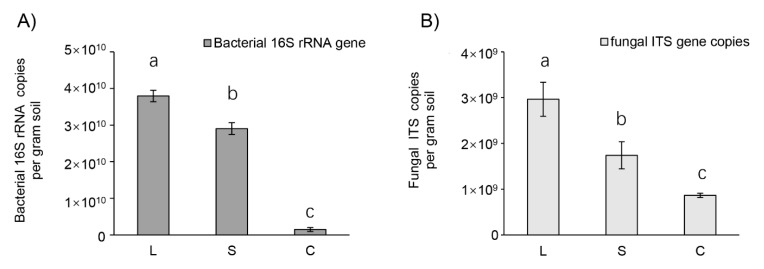
Microbial gene copy numbers in mushroom cultivation soils determined by qPCR. Values are given by primers targeting (**A**) bacterial 16S rRNA and (**B**) fungal ITS region in three different *Stropharia rugosoannulata* cultivation areas with various influencing intensities: C = control forest soil with no influence from mushroom cultivation; L = soil from furrows with low influence by mushroom cultivation; S = soil below fungal beds strongly influenced by mushroom cultivation. Different lowercase letters for bacterial or fungal gene copy numbers indicate significant differences (*p* < 0.05, ANOVA, Tukey HSD) of bacterial and fungal gene copies.

**Figure 7 jof-07-00775-f007:**
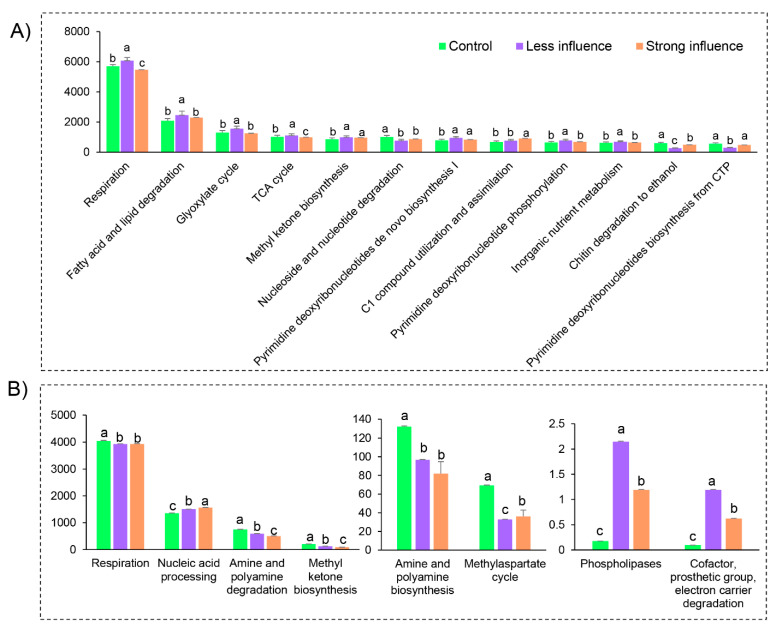
Microbial community functional abundance prediction based on MetaCyc genome database. Bacterial (**A**) and fungal (**B**) functional profiles in *Stropharia rugosoannulata* cultivation areas with different influencing intensities are shown as bar charts. Values are the mean from four bioreplicates, different lowercase letters indicate significant differences at *p* < 0.05 (ANOVA, Tukey HSD).

**Figure 8 jof-07-00775-f008:**
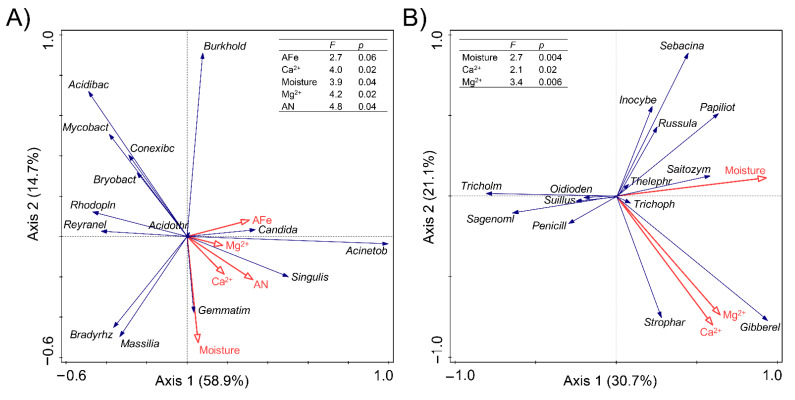
Relationships of the most influential soil properties and bacterial (**A**) and fungal (**B**) genera in soils with *Stropharia rugosoannulata* cultivation. Length of arrows represents the association strength of the respective soil properties with the microbial genera. Angle between vectors indicates the degree of their relationship (smaller angle means high correlation). Values on the axes illustrate the percentage explained by redundancy analysis (RDA) or canonical correspondence analysis (CCA), based on the gradient length by detrended correspondence analysis ordination axis. To reduce the large number of explanatory factors (soil properties, enzyme activities and microbial abundance) and avoid overfitting the RDA/CCA, the interactive selection of top three to five most influential variables was used. Details in CCA/RDA selection are explained in Section 2.5. Abbreviations used in the figure were described in Table 1 and Table 2.

**Figure 9 jof-07-00775-f009:**
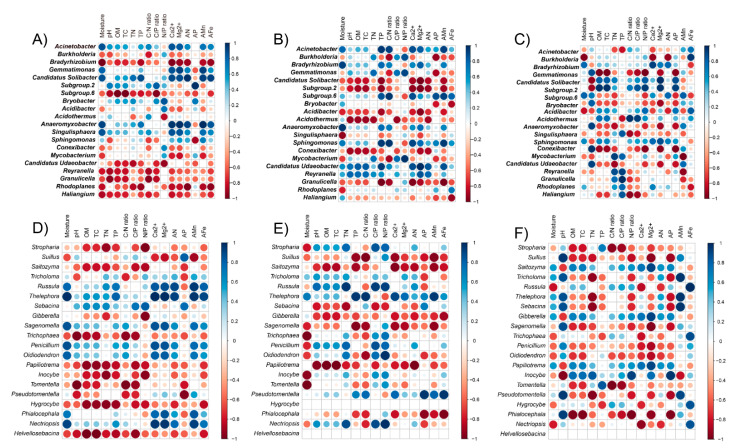
Spearman correlations between the dominant bacterial (**A**–**C**,**G**–**I**) and fungal taxa (**D**–**F**,**J**–**L**) and soil properties with low (**B**,**E**,**H**,**K**) and strong (**C**,**F**,**I**,**L**) influence of *Stropharia rugosoannulata* cultivation and control (**A**,**D**,**G**,**J**) soils, respectively. For clarity purposes, relationships between enzymatic activities and dominant microbial taxa are shown on the lower part of the figure (**G**–**L**). Size of circles in plot cells is proportional to correlation coefficients (Spearman’s *p*). Strength and direction of the correlations are denoted by circle size and color (as per scale bar). The scale bar extends from perfect correlation (dark blue, *r* = 1, dark red, *r* = −1). Large circles also denote a stronger correlation. Abbreviations of soil properties correspond to those described in Table 1 and Table 2.

**Figure 10 jof-07-00775-f010:**
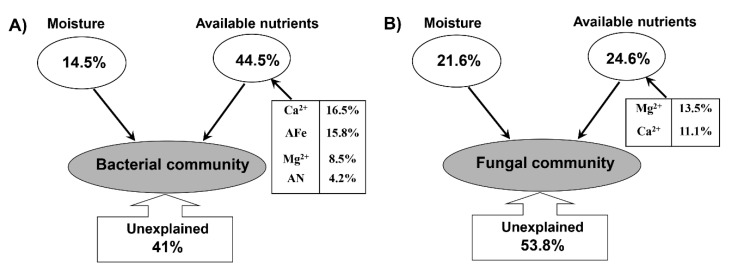
Variation partition analysis (VPA) of the effects of soil moisture and available nutrients on soil bacterial (**A**) and fungal (**B**) communities in *Stropharia rugosoannulata cultivation* areas.

**Table 1 jof-07-00775-t001:** Soil physical and chemical properties in *Stropharia rugosoannulata* cultivation plots.

Treatment	Moisture	pH	OM	TC	TN	TP	C:N	C:P	N:P
	(%)		(g kg^−1^)	(g kg^−1^)	(g kg^−1^)	(g kg^−1^)			
S	29.4(1.7) a	5.0(0.04) b	389(36) a	226(21) a	4.6(0.2) a	1.06(0.05) a	48.9(4.2) a	214(24) a	4.4(0.2) a
L	31.4(2.0) a	5.1(0.06) ab	76.5(11) b	44.4(6.7) b	2.4(0.4) b	0.95(0.10) a	18.8(1.9) b	46.9(5.2) b	2.5(0.4) b
C	20.8(0.2) b	5.2(0.01) a	74.4(15) b	43.2(8.7) b	2.5(0.4) b	0.88(0.09) ab	17.5(2.0) b	48.7(6.5) b	2.8(0.2) b

Values [means (SE), *n* = 4] followed by different letters are significantly different at *p* < 0.05 (ANOVA, Tukey HSD). Abbreviations: C = Control forest soil with no influence from mushroom cultivation; C:N, the ratio of total carbon to nitrogen; C:P, the ratio of total carbon to phosphorous; L = soil forest from furrows with low influence by mushroom cultivation; N:P, the ratio of total nitrogen to phosphorous; OM, organic matter; S = forest soil below fungal beds strongly influenced by mushroom cultivation; TC, total carbon; TN, total nitrogen; TP, total phosphorous.

**Table 2 jof-07-00775-t002:** Soil easily available nutrients in *Stropharia rugosoannulata* cultivation plots.

Treatment	Ca^2+^	Mg^2+^	AN	AP	AMn	AFe
	(mg kg^−1^)	(mg kg^−1^)	(mg kg^−1^)	(mg kg^−1^)	(mg kg^−1^)	(mg kg^−1^)
S	2934(399) a	1437(341) a	369(28) a	25(4.1) a	211(53) a	216(19) a
L	1017(122) b	80(32) b	232(33) b	6.5(1.7) b	29.3(15) b	38.1(5.9) b
C	973(121) b	74(27) b	225(44) b	5.2(0.8) c	24.3(12) b	33.1(7.4) b

Values (means (SE), *n* = 4) followed by different letters are significantly different at *p* < 0.05 (ANOVA, Tukey HSD). AN, alkaline hydrolysable nitrogen; AP, available phosphorous; AMn, available manganese; Afe, available Fe; ACP, acid phosphatase. Treatment abbreviations correspond to those shown in Table 1.

**Table 3 jof-07-00775-t003:** Soil enzyme activities in *Stropharia rugosoannulata* cultivation plots.

Treatment	ACP	Glucosidase	Cellobiosidase	Peroxidase
	(nmol g^−1^ h^−1^)	(nmol g^−1^ h^−1^)	(nmol g^−1^ h^−1^)	(nmol g^−1^ h^−1^)
S	1005(246) a	94.4(7.1) a	20.6(2.3) a	1613(12) a
L	746(480) b	77.5(6.8) b	14.1(2.0) b	1557(77) a
C	603(85) c	64.0(6.6) b	13.7(1.5) b	1510(121) a

Values (means (SE), *n* = 4) followed by different letters are significantly different at *p* < 0.05 (ANOVA, Tukey HSD). Treatment abbreviations correspond to those shown in Table 1.

**Table 4 jof-07-00775-t004:** The tests of significant value of Bray-Curtis distance matrix when comparing the influences of *Stropharia rugosoannulata* cultivation on soil bacterial and fungal communities.

	Group	Bacterial Community		Fungal Community	
		Pseudo-*F* ^b^	*p*-Value	Pseudo-*F*	*p*-Value
	Overall ^a^	1.509	0.032	2.903	0.002
PERMANOVA	C vs. L ^c^	1.743	0.036	4.821	0.031
	C vs. S	2.017	0.036	2.962	0.041
	L vs. S	0.842	0.546	1.469	0.140

^a^ Sample size = 12 (all groups) and 8 (pairwise results), permutations = 999; ^b^ Test statistic was pseudo-*F* for PERMANOVA results; ^c^ Treatment details are described in Table 1. Significant values (*p* < 0.05) are shown in bold.

## Data Availability

All sequence data have been deposited to the ENA Sequence Read Archive under accession number SRP264748.

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
