# Peer review of "Macrofungi Cultivation in Shady Forest Areas Significantly Increases Microbiome Diversity, Abundance and Functional Capacity in Soil Furrows"

_jof, 2021, doi:10.3390/jof7090775_

Round 1

Reviewer 1 Report

Dear authors, Dear Editor

I have read manuscript entitled “Macrofungi cultivation in forest shady area significantly increases microbiome network complexity and functional capacity in soils from inter-groove”. I think the topic is interesting and has high potential for applications. However, the work is strongly suffer from low replications (4) which eventually affect the validity and accuracy of results. There are some statistics and data analysis approaches that are suffer from such low replicate experiment. There are also possible errors for reported statistics values. At this round of review, I will focus on methodology and statistics.   

Specific comments/ suggestions

1) The authors should repeat experiments (for example add 4 more replicates) to maintain this manuscript structure. Otherwise, the authors have to cut all parts which are not supported by the experimental design. The authors have to think what to compensate the results that are cut. For example, if the authors have the soil store at -20 °C or -80 °C, the authors can do the soil enzyme activity analysis (hydrolytic and oxidative enzymes) and microbial biomass or abundances as determined by PLFA or qPCR.

2) The statistics and data analysis approaches which are not supported by the experimental design are for examples,  

2.1 Network analysis: the authors used the samples from all three treatments (all replicates) to make one network for bacteria and one network for fungi. Each treatment is very different from another, so the co-occurrences between microbes analysed by this way are not meaningful. They cannot be used to explain the different co-occurrence patterns among the three treatments. The best way is to repeat the experiment with 4 more replicates, then make 3 co-occurrence networks (from each treatment) for bacteria and fungi and then compare patterns of these three networks. Furthermore, authors can also do co-occurrence networks for bacteria-fungi interactions.

2.2 With 4 replicates, ANOSIM is not possible and not stable. This is clear from what  the authors reported in line 283  “For  the  bacterial  community,  NMDS  distinguished in  three  different mushroomcultivation areas (ANOSIM test, R = 0.313, P= 0.018), whereas the treatments between L and S were not significant (ANOSIM test, R = 0.591, P> 0.05)”. Normally, for ANOSIM when the R is higher than 0.25, we can interpret that there is some degree of separation of microbial communities but also with some degree of overlap. When the replicates are so small (as in this experiment), the P value is normally more than 0.05 (not significant). The uncertainty of ANOSIM may also indicate by the result in this study (when R = 0.313 it is significant but when R increased to 0.591, it is not significant). PERMANOVA may help, so please use it and remove the ANOSIM results.

2.3 With 4 replicates, the RDA will not work correctly. So, “Figure 8. Redundancy analysis (RDA) showing relationships of the three most influential physical and chemical soil properties and bacterial (A, C, E) and fungal (B, D, F) genera in soils with low and strong influence of Stropharia rugosoannulata cultivation and control soils. Length of arrows represents the association strength of the respective soil properties with the microbial genera. Angle between vectors indicates the degree of their relationship (smaller angle means high correlation).Values on the axes illustrate the percentage explained by RDA” is not valid at all. The authors fit all factors to 4 samples plot in ordination, this is not correct.

2.4 I suggest the authors to think which statistics and analysis can be used and which one cannot be used? Then, please only keep the ones that are valid.  

  1. “No-metric multi-dimensional scaling” does not exist. I think the authors mean “Non-metric multidimensional scaling (NMDS)”. If this is the case, please edit in the text.
  2. The authors did not explain the use of PERMANOVA in statistics section. The values reported in Table3 (t statistic) seem wrong, especially for C vs L and C vs S. The value (Pseudo F between 0.031 to 0.041 will never be significant). I think the authors put the P values there. The values for ANOSIM are also likely to be wrong. I recommend the authors to check all table and figures to present accurate information. Once I found such errors, I feel very disappointed and feel uncertain about what I am reading. The work from authors is interesting and nice but I also would like to read the correct data and description.

5) The authors can do fitting of data (NMDS with envfit), RDA, or CCA based on all treatments (put in one ordination (one for bacteria and one for fungi). The selection of method is depending on the data (please check the gradient length by DCA and select the correct method, if you have short gradient length then RDA if you have long gradient length then CCA). Please explain in statistics section. For this results, at least the authors can see which factors explain differences in microbial communities in these three treatments and also if such factors are the same between bacterial and fungal communities.

6) Please do PERMANOVA and/or variation partitioning to see which factors are the most important in explaining bacterial and fungal community composition.    

7) Please work on enzymes and microbial abundances (or biomass data). Please add them in introduction and set hypothesis. It will be nice to see if the treatments are actually affect the soil enzyme activities (especially oxidative enzymes due to presence of basidiomycetes Stropharia rugosoannulata). Please link enzymes and microbial abundances (or biomass data) with microbial richness and community analyses.

8) Please check “Stropharia spp” if they are “Stropharia rugosoannulata”. Please check top 50 fungal OTUs for their UNITE species hypothesis. Please put the data in supplementary information and mention briefly in the text. This will help authors to be more specific. For examples, line 20 “Compared to the control soil, mushroom cultivation formed distinct biomarkers in the soil below fungal beds (N-fixer Gematinonadet and Stropharia) and soil from inter-grooves  (Acidothermaces  and Venturia).”.  Which species of Stropharia and/or Venturia.

9) Figure S1 is interesting. Please consider move some important results to main text.

10) Bioinformatics are unclear. Please tell all the criterions for screening the high quality sequences (number of mismatch of primers, ambiguous nucleotide, ….). The authors have to report the sequencing depths for bacteria and fungi in bio informatics section. Please provide all sample rarefaction curves. I saw from the results section that that authors sequenced the majority of microbes, this is nice but you have to show it.

11) I am not native English, so I can’t help with language editing. I saw some language errors and wrong scientific terminology, please check  better and improve.

I wish the authors good luck for improving the manuscript.

Author Response

Dear authors, Dear Editor

I have read manuscript entitled “Macrofungi cultivation in forest shady area significantly increases microbiome network complexity and functional capacity in soils from inter-groove”. I think the topic is interesting and has high potential for applications. However, the work is strongly suffer from low replications (4) which eventually affect the validity and accuracy of results. There are some statistics and data analysis approaches that are suffer from such low replicate experiment. There are also possible errors for reported statistics values. At this round of review, I will focus on methodology and statistics.   

Specific comments/ suggestions

  • The authors should repeat experiments (for example add 4 more replicates) to maintain this manuscript structure. Otherwise, the authors have to cut all parts which are not supported by the experimental design. The authors have to think what to compensate the results that are cut. For example, if the authors have the soil store at -20 °C or -80 °C, the authors can do the soil enzyme activity analysis (hydrolytic and oxidative enzymes) and microbial biomass or abundances as determined by PLFA or qPCR.

We strongly agree with the reviewer`s opinion in relation to the fact that a higher number of replicates would produce a deeper understanding of the soil and microbial changes originated by the cultivation of Stropharia rugosoannulata. Yet the time and resources available to conduct the study did not allow to consider a more extensive sampling in a thorough but seminal study. We appreciate the comment and have gained in more fully recognizing the need for such an approach in subsequent works.

We deeply appreciate the kind advices and valuable time of the reviewer in order to provide us alternatives to strength our study with the number of replicates that we have sampled. Then, based on the fact that we have stored back up samples of all soil replicates and following the reviewer suggestions, we have: i) evaluated the enzymes glucosidase, cellobiosidase and peroxidases in all replicates; ii) performed a quantitative real-time PCR to quantify gene copy numbers of soil bacteria and fungi in order to understand the changes of microbial abundances in the different treatments; iii) cut all parts which are not supported by the experimental design; iv) Remade the figure 8, presenting a general analysis in the studied cultivation system for both bacteria and fungi and its relationship with the main chemical and physical soil properties; v) added a Spearman analysis which shows the correlations between the newly evaluated enzymes and the main bacterial and fungal groups in the different evaluated microniches to strength our research; vi) cut all parts which are not supported by the experimental design; and vii) carefully reviewed the whole manuscript, improving the writing-up of the text. We hope our effort have enhanced the overall scientific standard of the updated manuscript we are re-submitting. We would like to highlight that we are very grateful to the reviewer`s valuable comments, suggestions and corrections.    

2) The statistics and data analysis approaches which are not supported by the experimental design are for examples,  

2.1 Network analysis: the authors used the samples from all three treatments (all replicates) to make one network for bacteria and one network for fungi. Each treatment is very different from another, so the co-occurrences between microbes analysed by this way are not meaningful. They cannot be used to explain the different co-occurrence patterns among the three treatments. The best way is to repeat the experiment with 4 more replicates, then make 3 co-occurrence networks (from each treatment) for bacteria and fungi and then compare patterns of these three networks. Furthermore, authors can also do co-occurrence networks for bacteria-fungi interactions.

We strongly agree with the reviewer`s comments, therefore due to the limited value added to the whole story of network analysis mixing samples from all replicates, including the three treatments, we have removed this analysis in the updated manuscript.

2.2 With 4 replicates, ANOSIM is not possible and not stable. This is clear from what the authors reported in line 283 “For the bacterial community, NMDS distinguished in three different mushroom cultivation areas (ANOSIM test, R = 0.313, P= 0.018), whereas the treatments between L and S were not significant (ANOSIM test, R = 0.591, P> 0.05)”. Normally, for ANOSIM when the R is higher than 0.25, we can interpret that there is some degree of separation of microbial communities but also with some degree of overlap. When the replicates are so small (as in this experiment), the P value is normally more than 0.05 (not significant). The uncertainty of ANOSIM may also indicate by the result in this study (when R = 0.313 it is significant but when R increased to 0.591, it is not significant). PERMANOVA may help, so please use it and remove the ANOSIM results.

We strongly agree with the recommendation, thus following the valuable suggestion and the proposed alternative analysis, we have removed the ANOSIM results and conducted PERMANOVA analysis, which are now presented in both the methods and result sections, particularly in the updated 3.4 subsection of results.  

2.3 With 4 replicates, the RDA will not work correctly. So, “Figure 8. Redundancy analysis (RDA) showing relationships of the three most influential physical and chemical soil properties and bacterial (A, C, E) and fungal (B, D, F) genera in soils with low and strong influence of Stropharia rugosoannulata cultivation and control soils. Length of arrows represents the association strength of the respective soil properties with the microbial genera. Angle between vectors indicates the degree of their relationship (smaller angle means high correlation).Values on the axes illustrate the percentage explained by RDA” is not valid at all. The authors fit all factors to 4 samples plot in ordination, this is not correct.

Based on the reviewer’s suggestion, we have modified the previous Figure 8. In order to compensate this removal and to contribute to the understanding of the relationships between soil properties and bacterial and fungal genera in the different evaluated microniches, we have added a Spearman analysis of the four newly evaluated soil enzymes and the most abundant bacterial and fungal genera. This is presented now as Figure 10, in addition to the previous Spearman analysis presented in the current Figure 9. Interestingly, the new analysis showed that the highest enzyme production for most of the bacterial and fungal genera was recorded in the furrows. This analysis strength the idea that this soil microniche is an important functional hotspot, as shown by most of the conducted analysis in the whole manuscript.     

2.4 I suggest the authors to think which statistics and analysis can be used and which one cannot be used? Then, please only keep the ones that are valid.  

We are very grateful for the reviewer’s suggestion, then we have kept only the statistics and analysis which are valid and we have also thought how to compensate the analysis which have been cut.

  1. “No-metric multi-dimensional scaling” does not exist. I think the authors mean “Non-metric multidimensional scaling (NMDS)”. If this is the case, please edit in the text.

Thank you for your valuable correction. We have made the correction throughout the manuscript.

  1. The authors did not explain the use of PERMANOVA in statistics section. The values reported in Table3 (t statistic) seem wrong, especially for C vs L and C vs S. The value (Pseudo F between 0.031 to 0.041 will never be significant). I think the authors put the P values there. The values for ANOSIM are also likely to be wrong. I recommend the authors to check all table and figures to present accurate information. Once I found such errors, I feel very disappointed and feel uncertain about what I am reading. The work from authors is interesting and nice but I also would like to read the correct data and description.

We have included the explanation for the use of PERMANOVA in the Material and Methods section specifically in the 2.5 Statistics section. We sincerely apologize for our previous mistakes shown in Table 3. In the enhanced version of our manuscript, we have carefully re-worked PERMANOVA and included now the correct values. Taking into account the low replicate number we have removed ANOSIM results from Table 3. In addition, we would like to express our gratitude for the reviewer’ comments, and carefully checked the information and analysis in all tables and figures of the enhanced manuscript.

5) The authors can do fitting of data (NMDS with envfit), RDA, or CCA based on all treatments (put in one ordination (one for bacteria and one for fungi). The selection of method is depending on the data (please check the gradient length by DCA and select the correct method, if you have short gradient length then RDA if you have long gradient length then CCA). Please explain in statistics section. For this results, at least the authors can see which factors explain differences in microbial communities in these three treatments and also if such factors are the same between bacterial and fungal communities.

We deeply appreciate the valuable suggestion of the reviewer. Then, following this valuable recommendation, we remade the previous analysis, and included in the Results section an analysis based on all treatments, one for bacteria and one for fungi, of the main soil factors which could explain the differences in microbial communities in our study system and if such factors were the same for bacteria and fungal assemblies. This analysis is currently shown in the Figure 8 of the enhanced manuscript. We also wrote a detailed explanation of the RDA/CCA analysis in the Materials and Methods section, in the 2.5 Statistical analysis subsection.    

6) Please do PERMANOVA and/or variation partitioning to see which factors are the most important in explaining bacterial and fungal community composition.    

Thank you for your valuable suggestion, we have conducted PERMANOVA and also, we have added a new Figure showing the results of variation partitioning analysis (VPA) as Figure 10, explaining as well in detail the followed procedures in the Material and methods section.

7) Please work on enzymes and microbial abundances (or biomass data). Please add them in introduction and set hypothesis. It will be nice to see if the treatments are actually affect the soil enzyme activities (especially oxidative enzymes due to presence of basidiomycetes Stropharia rugosoannulata). Please link enzymes and microbial abundances (or biomass data) with microbial richness and community analyses. 

Following the suggestion by the reviewer, we have measured the activities of two hydrolase (βGlucosidase, and α-D-1, 4-Cellobiosidase) and one oxidase (Peroxidase). Also, for quantifying gene copy numbers of soil bacteria and fungi, a quantitative real-time PCR was conducted which shows the changes in microbial abundances in the updated manuscript. We have added this analysis in the Introduction by updating the hypothesis 1 (H1) accordingly. Additionally, we conducted Spearman relationships between the evaluated enzymes and the bacterial and fungal genera with the highest abundance; and included this analysis as part of the Figure 9 (G-L). Therefore, we have enriched the Material and Methods, Results and Discussion sections by including the procedures and interpretation of this analysis.

8) Please check “Stropharia spp” if they are “Stropharia rugosoannulata”. Please check top 50 fungal OTUs for their UNITE species hypothesis. Please put the data in supplementary information and mention briefly in the text. This will help authors to be more specific. For examples, line 20 “Compared to the control soil, mushroom cultivation formed distinct biomarkers in the soil below fungal beds (N-fixer Gematinonadet and Stropharia) and soil from inter-grooves (Acidothermaces and Venturia).” Which species of Stropharia and/or Venturia.

Yes, they are “Stropharia rugosoannulata”. Therefore, as suggested by the reviewer, we have checked the top 50 fungal OTUs for their UNITE species hypothesis and the information has been included as a supplementary table. In the case of the summary as the instructions for authors indicate that this should be no longer than 200 words, we have re-written it, in order to fulfill this request.

9) Figure S1 is interesting. Please consider move some important results to main text.

Thank you for your valuable suggestion. Therefore, we have included this figure as part of the main text.

10) Bioinformatics are unclear. Please tell all the criterions for screening the high quality sequences (number of mismatch of primers, ambiguous nucleotide). The authors have to report the sequencing depths for bacteria and fungi in bio informatics section. Please provide all sample rarefaction curves. I saw from the results section that that authors sequenced the majority of microbes, this is nice but you have to show it.

Bioinformatic details have been now included in the Material and Methods section, 2.3 subsection. Additionally, based on the reviewer’s suggestion, we have included a supplementary figure 1 to provide all sample rarefaction curves. Finally in the subsection 3.2 a paragraph explaining rarefraction details was added.

11) I am not native English, so I can’t help with language editing. I saw some language errors and wrong scientific terminology, please check better and improve.

To address the English language quality and clarity of the manuscript, we have carefully checked the whole text. With these careful inputs we hope we have now enhanced both the overall scientific quality and English standard of the manuscript.

I wish the authors good luck for improving the manuscript.

Reviewer 2 Report

Line 18. Should be “Non-metric” or “Nonmetric”, not “No-metric”

Line 21. I think “Gemmatimonadetes” is meant.

Line 85. “The region is subtropical climate”. I think you mean “The region is characterized by a subtropical climate”

Lines 103–104. “The rest samples”? I think you mean “The rest of the samples”  or “The remainder of the samples”

Line 106. “in a soil to water mixtures” should either be “in a soil to water mixture” or “in soil to water mixtures”

Lines 158–168. The citation to Reference [34] should appear somewhere here but seems to be  missing.

Line 206. “followed different letters” should read “followed by different letters”. Same correction on line 212.

Line 239–243. This looks like one sentence, but doesn’t make sense. Perhaps some punctuation is needed, for example; splitting it into two sentences.

Line 240. “the rest 21 genera” should read “the rest of the 21 genera” or “the remainder of the 21 genera”

Line 270. “did not differ among in different” doesn’t make sense. Needs rewriting.

Line 283. Do you mean “distinguished the three different mushroom cultivation areas”?

Lines 284–285. Replace by “whereas the difference between treatments L and S was not significant”

Line 286. “enlarged”? Do you mean “increased”?

Line 327. “so was for”? Did you mean “as was the case for”? Not clear; it needs rewriting.

Line 332. Delete “(Table 4)”. The citation on the following line covers both.

Line 350. Insert “the” before “top” and delete “that” before “extracted”

Line 351. Insert “the” before “overall”

Line 357. Insert “the” before “bacterial”

Line 364. Insert “the” before “fungal”

Line 394. Fig. 8. The axes should be labelled. Presumably, the horizontal axis is RDA1 and the vertical axis is RDA2. The caption says that the values on the axes illustrate the percentage explained by RDA. But the values are not percentages, as the values shown there range between -1 and +1.

Line 418. “significant” should read “significantly”

Line 460. Delete “In” and start sentence with “Consistent”

Line 471. “in term of” should read “in terms of” and “compared” should replace “comparing”

Lines 498–500. This sentence doesn’t make sense. How can “varied soil driving factors” be “an importance factor”? Rewrite.

Lines 501–503. This seems to be a sentence fragment; it doesn’t make sense as it stands.

Author Response

Reviewer 2#

Line 18. Should be “Non-metric” or “Nonmetric”, not “No-metric”

Correction has been made.

Line 21. I think “Gemmatimonadetes” is meant.

As the sentence was not accurate neither clear, it has been modified.

Line 85. “The region is subtropical climate”. I think you mean “The region is characterized by a subtropical climate”

Change has been made.

Lines 103–104. “The rest samples”? I think you mean “The rest of the samples”  or “The remainder of the samples”

Correction has been made.

Line 106. “in a soil to water mixtures” should either be “in a soil to water mixture” or “in soil to water mixtures”

Suggested change has been made.

Lines 158–168. The citation to Reference [34] should appear somewhere here but seems to be  missing.

We have included Segata et al. (2011) as reference [34].

Line 206. “followed different letters” should read “followed by different letters”. Same correction on line 212.

Suggested changes have been made.

Line 239–243. This looks like one sentence, but doesn’t make sense. Perhaps some punctuation is needed, for example; splitting it into two sentences.

Following your valuable suggestion, the original sentence has been split into two sentences.

Line 240. “the rest 21 genera” should read “the rest of the 21 genera” or “the remainder of the 21 genera”

Suggested correction has been made.

Line 270. “did not differ among in different” doesn’t make sense. Needs rewriting.

The sentence has been rewritten.

Line 283. Do you mean “distinguished the three different mushroom cultivation areas”?

Yes. We have changed here as suggested by the reviewer.

Lines 284–285. Replace by “whereas the difference between treatments L and S was not significant”

Suggested change has been made.

Line 286. “enlarged”? Do you mean “increased”?

Suggested change has been made.

Line 327. “so was for”? Did you mean “as was the case for”? Not clear; it needs rewriting.

Based on the comments of the first reviewer, we realized that the network part was not quite relevant. Therefore, the whole subsection “network of bacterial and fungal community” has been removed.

Line 332. Delete “(Table 4)”. The citation on the following line covers both.

The Table 4 has been deleted.

Line 350. Insert “the” before “top” and delete “that” before “extracted”

Suggested changes have been made.

Line 351. Insert “the” before “overall”

Suggested change has been made.

Line 357. Insert “the” before “bacterial”

Suggested change has been made.

Line 364. Insert “the” before “fungal”

Suggested change has been made.

Line 394. Fig. 8. The axes should be labelled. Presumably, the horizontal axis is RDA1 and the vertical axis is RDA2. The caption says that the values on the axes illustrate the percentage explained by RDA. But the values are not percentages, as the values shown there range between -1 and +1.

We have reworked and labelled the axes of the Fig. 8.

Line 418. “significant” should read “significantly”

Suggested change has been made.

Line 460. Delete “In” and start sentence with “Consistent”

Suggested change has been made.

Line 471. “in term of” should read “in terms of” and “compared” should replace “comparing”

Suggested changes have been made.

Lines 498–500. This sentence doesn’t make sense. How can “varied soil driving factors” be “an importance factor”? Rewrite

Following the valuable suggestion, the sentence has been deleted.

Lines 501–503. This seems to be a sentence fragment; it doesn’t make sense as it stands.

Paragraph has been rewritten.

Round 2

Reviewer 1 Report

Dear Authors and Editor,

Apart from the weak experimental design (which cannot be changed), other critical comments have been sufficiently addressed. The authors put nice efforts for qPCR and enzymes which help to increase the quality of this paper. I have some few comments with statistics and formatting that the authors must take care and read better. Please see specific comments below.

1) Please give the protocols for all chemical analysis in supplementary information and cite them in main text. For enzyme activities, please explain briefly the kit and how to prepare the samples.

2) Take care of typo and format. For examples, Line 17, 20,: P value should be written as Italic; Line 125 check H2SO4 , the number 2 should be subscript. The authors please check for all these format issues. It will be not so good that at the end there will be errors due to the formatting. P values in text and figures should be consistent (now they are written both with P or p and not Italic, please check). All statistic values should also be Italic, such as “r”, please check.   

3) The authors should add the “rarified observed richness” and describe in the results.  

4) Figure 7b, c, d. Microbial community functional abundance prediction based on MetaCyc genome database, please compared all three treatments (put different letters for differences like a, b, c). Please add SE to the bars. For some bars, the values are so less and we see nothing. Please solve this problem.

5) For Table and figures, the authors should bold the significant P values (both P and the numbers), so it is easy to recognize (marginal significant can be in Italic).

6) Figure 3, please put different letters for differences like a, b, c. Please check for normality and equality of variances before using ANOVA. The authors can try to transform data to meet the criteria for parametric test too. If the authors cannot use ANOVA, can try KW test. Please see Chao1 for fungal richness, the P value was significant but no difference was detected for pair-wise comparison. The variances among different treatments look clearly dissimilar.   

7) For Bioinformatics, what did the authors do with singletons?

8) Figure 6, please separately plot bacteria and fungi as the values are very much different (we see almost nothing for bacteria).

9) Please do final check for numbers in texts and figures. I wish the authors good luck with revision.

Author Response

Dear Authors and Editor,

Apart from the weak experimental design (which cannot be changed), other critical comments have been sufficiently addressed. The authors put nice efforts for qPCR and enzymes which help to increase the quality of this paper. I have some few comments with statistics and formatting that the authors must take care and read better. Please see specific comments below.

1) Please give the protocols for all chemical analysis in supplementary information and cite them in main text. For enzyme activities, please explain briefly the kit and how to prepare the samples.

As kindly suggested, we have described the protocols used for all chemical analysis. Additionally, in the case of the enzymatic activities we have briefly explained the used kits and the methods conducted to measure the four evaluated soil enzymes. This information has been included in the new supplementary file 1 and cited this file in the main text.

2) Take care of typo and format. For examples, Line 17, 20,: P value should be written as Italic; Line 125 check H2SO4 , the number 2 should be subscript. The authors please check for all these format issues. It will be not so good that at the end there will be errors due to the formatting. P values in text and figures should be consistent (now they are written both with P or p and not Italic, please check). All statistic values should also be Italic, such as “r”, please check.  

Thank you for your kind suggestion. We have now carefully checked the typo and format. Suggested corrections in lines 17 and 125 have been made. Additionally, all statistic values have been written in italic and the nonstandard writing of “Ca2+” and “Mg2+” in Figures 8 and 10 have also been changed.

3) The authors should add the “rarified observed richness” and describe in the results. 

We have added the rarified observed richness in the supplementary figure 1, and described it in detail for both the bacteria and fungi in the subsection 3.2

4) Figure 7b, c, d. Microbial community functional abundance prediction based on MetaCyc genome database, please compared all three treatments (put different letters for differences like a, b, c). Please add SE to the bars. For some bars, the values are so less and we see nothing. Please solve this problem.

Thank you for your valuable suggestion. Figure 7 has been remade, we have calculated the statistic differences among the three evaluated treatments, included different lower-case letters when there were significant differences and added SE to the bars. Additionally, in order to have a more didactic presentation, in the case of fungal functional abundance prediction values we have separated them in three different groups taking into account the scale differences.  

5) For Table and figures, the authors should bold the significant P values (both P and the numbers), so it is easy to recognize (marginal significant can be in Italic).

Following this valuable suggestion, we have written in bold type the significant p values in Table 3.

6) Figure 3, please put different letters for differences like a, b, c. Please check for normality and equality of variances before using ANOVA. The authors can try to transform data to meet the criteria for parametric test too. If the authors cannot use ANOVA, can try KW test. Please see Chao1 for fungal richness, the P value was significant but no difference was detected for pair-wise comparison. The variances among different treatments look clearly dissimilar.

As suggested by the reviewer, in the case of data presented in Figure 3, we have checked the normality and equality of variances, and added different letters when there were significant differences at p = 0.05 (ANOVA) between means (Tukey’s HSD pairwise comparisons, n = 4). The statistic results have now been included in Figure 3.

7) For Bioinformatics, what did the authors do with singletons?

The singletons (sequences that occurred only once in dataset) were removed from downstream analyses, with the aim of improving sequencing accuracy and avoid overestimation of bacterial diversity.

8) Figure 6, please separately plot bacteria and fungi as the values are very much different (we see almost nothing for bacteria).

Thank you for the valuable suggestion, therefore we have remade figure 6.

9) Please do final check for numbers in texts and figures. I wish the authors good luck with revision.

The numbers in texts and figures have carefully been checked. In order to facilitate the review of the updated version, we have highlighted in gray color the text which has been modified. We are deeply grateful towards your valuable and constructive comments and suggestions during the reviewing process. They have strongly enhanced our manuscript, thank you very much.